

# A quasi-one-dimensional ice mélange flow model based on continuum descriptions of granular materials

Jason M. Amundson[1], Alexander A. Robel[2], Justin C. Burton[3], and Kavinda Nissanka[3]

[1]Department of Natural Sciences, University of Alaska Southeast, Juneau, AK, USA
[2]School of Earth and Atmospheric Sciences, Georgia Institute of Technology, Atlanta, GA, USA
[3]Department of Physics, Emory University, Atlanta, GA, USA

**Correspondence:** Jason Amundson (jmamundson@alaska.edu)

**Abstract.** Field and remote sensing studies suggest that ice mélange influences glacier-fjord systems by exerting stresses on glacier termini and releasing large amounts of freshwater into fjords. The broader impacts of ice mélange over long time scales are unknown, in part due to a lack of suitable ice mélange flow models. Previous efforts have included modifying existing viscous ice shelf models, despite the fact that ice mélange is fundamentally a granular material, and running computationally
expensive discrete element simulations. Here, we draw on laboratory studies of granular materials, which exhibit viscous flow when stresses greatly exceed the yield point, plug flow when the stresses approach the yield point, and stress transfer via force chains. By implementing the nonlocal granular fluidity rheology into a depth- and width-integrated stress balance equation, we produce a numerical model of ice mélange flow that is consistent with our understanding of well-packed granular materials and that is suitable for long time-scale simulations. For parallel-sided fjords, the model exhibits two possible steady state solutions.
When there is no calving of new icebergs or melting of previously calved icebergs, the ice mélange is pushed down fjord by the advancing glacier terminus, the velocity is constant along the length of the fjord, and the thickness profile is exponential. When calving and melting are included, the ice mélange evolves to another steady state in which its location is fixed relative to the fjord walls, the thickness profile is relatively steep, and the flow is extensional. For the latter case, the model predicts that the steady-state ice mélange buttressing force depends on the surface and basal melt rates through an inverse power law
relationship, decays roughly exponentially with both fjord width and gradient in fjord width, and increases with the iceberg calving flux. The increase in buttressing force with the calving flux, which depends on glacier thickness, appears to occur more rapidly than the force required to prevent the capsize of full-glacier-thickness icebergs, suggesting that glaciers with high calving fluxes may be more strongly influenced by ice mélange than those with small fluxes.

# 1 Introduction

Ice mélange is an intrinsically granular material that is comprised of icebergs, brash ice, and sea ice packed together at the ocean surface. In some fjords, where iceberg productivity is high, ice mélange can persist year round. In others, it forms for



a few months in winter, when sea ice binds the iceberg clasts together, and then breaks apart each spring. Ice mélange is a highly heterogeneous material, with clast dimensions varying from meters to hundred of meters in both horizontal and vertical
dimensions. The large vertical dimension of ice mélange suggests that some processes that are important for sea ice and river ice, such as ridging and rafting, are likely unimportant for the flow of ice mélange.

Previous work has established that glacier advance between major calving events can result in the formation of an ice mélange wedge that flows quasi-statically and that exerts a force per unit width on the glacier terminus on the order of $10^7$ N m$^{-1}$ (Robel, 2017; Burton et al., 2018; Amundson and Burton, 2018). This load may be sufficient to inhibit calv-
ing and capsizing of new icebergs (e.g., Amundson et al., 2010; Krug et al., 2015; Bassis et al., 2021; Crawford et al., 2021; Schlemm et al., 2022), which is supported by studies that have linked break-up of a seasonal ice mélange wedge to the onset of calving in early summer (Cassotto et al., 2015; Bevan et al., 2019; Xie et al., 2019; Joughin et al., 2020). In locations where ice mélange persists year round, it appears to remain sufficiently strong to influence the timing and seasonality of calving events (Wehrlé et al., 2023). Terrestrial radar data indicates that ice mélange flow becomes incoherent at the grain scale in the hours
preceding major calving events (Cassotto et al., 2021), suggesting a weakening of the ice mélange, and that dynamic jamming occurs once an iceberg calves into the fjord (Peters et al., 2015).

Recent work has demonstrated that icebergs are also important sources of freshwater in fjords (Enderlin et al., 2016, 2018; Moon et al., 2017; Mortensen et al., 2020), especially during winter, and that this distributed release of freshwater has implications for fjord circulation and submarine melting of glacier termini. The presence of icebergs tends to freshen and cool fjords,
but also helps to enhance estuarine circulation and drive warm water into fjords, where it comes into contact with and melts glacier termini (Davison et al., 2020). Icebergs additionally create complex flow pathways and tend to decrease the velocity of subsurface waters (Hughes, 2022).

The conclusion of many studies is that there is a strong need for an ice mélange model that is consistent with its granular nature and that can be mechanically and thermodynamically coupled to the glacier-ocean system. Previous modeling attempts
have used discrete element models (Robel, 2017; Burton et al., 2018), modified existing ice shelf models (Pollard et al., 2018), incorporated sparse icebergs into sea ice models (Vaňková and Holland, 2017; Kahl et al., 2023), or used simple parameterizations (Schlemm and Levermann, 2021). Here we develop a depth-integrated ice mélange flow model that uses the nonlocal granular fluidity rheology (Henann and Kamrin, 2013), which has been developed from experiments of granular materials and that has successfully described a variety of granular flows. In order to investigate the basic behavior of the model and to
expedite development of coupled glacier-ocean-mélange models, we convert the model into a quasi-one-dimensional model by separately parameterizing the longitudinal and shear stresses and then integrating across the fjord. This approach closely mimics one that is commonly used for developing flow line models for ice shelves, as does the numerical implementation of the model. Thus, this study provides a framework by which realistic models of ice mélange can be incorporated into coupled glacier-ocean models.



## 2 Model description

### 2.1 Depth-integrated flow equations

We start by defining the strain rate and effective strain rate as $\dot{\varepsilon}_{ij} = 1/2(\partial u_i/\partial x_j + \partial u_j/\partial x_i)$ and $\dot{\varepsilon}_e = (\dot{\varepsilon}_{ij}\dot{\varepsilon}_{ij}/2)^{1/2}$, where $u_i = \langle u, v, w \rangle$ and $x_i = \langle x, y, z \rangle$ are the velocity and position vectors. We use the Cauchy stress tensor, $\sigma_{ij} = \sigma_{ji}$, with the convention that positive stresses are extensional. We partition the stress tensor into tectonic stresses $R_{ij}$ and the granular static

pressure $\tilde{p}$ by setting $\sigma_{ij} = R_{ij} - \tilde{p}\delta_{ij}$, where $\delta_{ij}$ is the Kronecker delta. We assume that the ice mélange is tightly packed and incompressible ($\dot{\varepsilon}_{kk} = 0$), at flotation, and evolving slowly enough that acceleration can be neglected. Consequently, the model is best suited for simulating ice mélange behavior in fjords where it persists year round or for winter conditions in fjords where it forms seasonally. Further modifications would be required to model rapid flow associated with calving events or complete disintegration of ice mélange in summer. For well-packed ice mélange the inertial number is typically very small ($< 10^{-5}$; see

Amundson and Burton, 2018), which places it well within the quasi-static regime. Under steady flow conditions the equations of motion are then $\partial\sigma_{ij}/\partial x_j = \rho g_{\text{eff}}\delta_{iz}$, where $\rho$ is the material density and

$$g_{\text{eff}} = \begin{cases} g & z \geq 0 \\ -\left(1 - \frac{\rho}{\rho_w}\right)g & z < 0, \end{cases} \tag{1}$$

with $\rho_w$ the density of water, $g$ the gravitational acceleration, and $z = 0$ corresponding to sea level. This formulation differs from that used to derive the shallow shelf approximation (SSA) that is used for modeling ice shelves because seawater fills

void spaces within ice mélange and thus the static pressure does not depend solely on the weight of the overlying ice. The static pressure is found by integrating Equation (1):

$$\tilde{p}(z) = \begin{cases} \rho g\left[\left(1 - \frac{\rho}{\rho_w}\right)H - z\right] & z \geq 0 \\ \rho g\left(1 - \frac{\rho}{\rho_w}\right)\left(H + \frac{\rho_w}{\rho}z\right) & z < 0, \end{cases} \tag{2}$$

where $H$ is the ice mélange thickness.

In addition, we assume that basal shear stresses are small and therefore vertical shear is negligible; consequently velocities,

strain rates, and stresses do not vary with depth. Thus, after partitioning the stress tensor, vertical integration of the momentum equations leads to

$$\frac{\partial}{\partial x}(HR_{xx}) + H\frac{\partial}{\partial y}R_{xy} = 2H\frac{\partial\tilde{P}}{\partial x}$$
$$\frac{\partial}{\partial y}(HR_{yy}) + H\frac{\partial}{\partial x}R_{xy} = 2H\frac{\partial\tilde{P}}{\partial y}, \tag{3}$$

where $R_{ij}$ now refers to depth-averaged values and

$$\tilde{P} = \frac{1}{2}\rho g\left(1 - \frac{\rho}{\rho_w}\right)H \tag{4}$$





is the depth-averaged granular static pressure. This continuum description of ice mélange will produce a smooth basal surface; with such geometries we anticipate very small shear stresses along the base. Hughes (2022) modeled the flow of water through and beneath rough ice mélange and found that the drag force per unit width is on the order of $10\,\mathrm{kN\,m^{-1}}$, which is about two orders of magnitude smaller than the drag forces due to lateral shear that we calculate in our model. We therefore neglect basal shear stresses but note that future efforts may need to include them in order to model fjords in which ice mélange does not remain well packed or persist year round.

The depth-averaged deviatoric stress is defined as $\sigma'_{ij} = \sigma_{ij} - P\delta_{ij}$, where $P = (\sigma_{xx} + \sigma_{yy} + \sigma_{zz})/3$ is the depth-averaged isometric pressure. By vertically integrating the tectonic stress and comparing the result to the deviatoric stress, we find that

$$R_{ij} = \sigma'_{ij} - \left(P - \tilde{P}\right)\delta_{ij}. \tag{5}$$

When $i = j = z$, Equation (5) can be re-written to show that

$$\left(P - \tilde{P}\right) = \sigma'_{zz} - R_{zz} = -\sigma'_{xx} - \sigma'_{yy} - R_{zz}. \tag{6}$$

Due to its granular nature ice mélange will not flex like ice shelves, which are often close to being in hydrostatic equilibrium except near their grounding lines. We therefore assume that bridging effects are negligible (i.e., that the weight of the ice mélange is locally supported by seawater) and therefore $R_{zz} = 0$. Thus Equation (5) becomes

$$R_{ij} = \sigma'_{ij} + (\sigma'_{xx} + \sigma'_{yy})\delta_{ij}. \tag{7}$$

Following Amundson and Burton (2018), we assume a depth-integrated viscoplastic rheology for granular materials:

$$\sigma'_{ij} = \frac{\mu P}{\dot{\epsilon}_e}\dot{\epsilon}_{ij}, \tag{8}$$

where $\mu$ is an effective coefficient of friction within the ice mélange that depends nonlinearly on the strain rate (see below). From Equation (6) we see that $P = \tilde{P} - \sigma'_{xx} - \sigma'_{yy}$ and therefore

$$\sigma'_{ij} = \frac{\mu(\tilde{P} - \sigma'_{xx} - \sigma'_{yy})}{\dot{\epsilon}_e}\dot{\epsilon}_{ij}. \tag{9}$$

Solving for $\sigma'_{xx}$ and $\sigma'_{yy}$, plugging the results back into Equation (9), and rearranging yields

$$\sigma'_{ij} = \frac{\mu\tilde{P}}{\dot{\epsilon}_e + \mu(\dot{\epsilon}_{xx} + \dot{\epsilon}_{yy})}\dot{\epsilon}_{ij}. \tag{10}$$

The resistive stress is then found by inserting Equation (10) into Equation (7):

$$R_{ij} = \frac{\mu\tilde{P}}{\dot{\epsilon}_e + \mu(\dot{\epsilon}_{xx} + \dot{\epsilon}_{yy})}\left[\dot{\epsilon}_{ij} + (\dot{\epsilon}_{xx} + \dot{\epsilon}_{yy})\delta_{ij}\right]. \tag{11}$$

Substituting Equation (4) into Equation (11) and the result into Equation (3), dividing by $\rho g(1 - \rho/\rho_w)/2$, and rearranging, gives

$$\frac{\partial}{\partial x}\left[\frac{\mu H^2}{\dot{\epsilon}_e + \mu(\dot{\epsilon}_{xx} + \dot{\epsilon}_{yy})}(2\dot{\epsilon}_{xx} + \dot{\epsilon}_{yy})\right] + H\frac{\partial}{\partial y}\left[\frac{\mu H}{\dot{\epsilon}_e + \mu(\dot{\epsilon}_{xx} + \dot{\epsilon}_{yy})}\dot{\epsilon}_{xy}\right] = 2H\frac{\partial H}{\partial x}$$

$$\frac{\partial}{\partial y}\left[\frac{\mu H^2}{\dot{\epsilon}_e + \mu(\dot{\epsilon}_{xx} + \dot{\epsilon}_{yy})}(2\dot{\epsilon}_{yy} + \dot{\epsilon}_{xx})\right] + H\frac{\partial}{\partial x}\left[\frac{\mu H}{\dot{\epsilon}_e + \mu(\dot{\epsilon}_{xx} + \dot{\epsilon}_{yy})}\dot{\epsilon}_{xy}\right] = 2H\frac{\partial H}{\partial y}. \tag{12}$$



In viscoplastic granular rheologies, $\mu$ is a complex function of $\dot{\epsilon}_e$. We adopt the nonlocal granular fluidity rheology of Henann and Kamrin (2013), which is derived from laboratory experiments that demonstrate viscous flow at high stress and plug flow at low stress. The rheology is nonlocal because it enables mesoscopic regions of yielding to cause elastic deformation in adjacent jammed regions, and it is particularly well suited for ice mélange because it has been developed from experiments of flows associated with low inertial numbers. The nonlocal granular fluidity rheology has successfully modeled a variety of granular flows, including flow down a rough plane (Kamrin and Henann, 2015), creep of intruders in low stress regions (Henann and Kamrin, 2014), annular shear with various grain geometries and materials (Fazelpour et al., 2022), and silo clogging (Dunatunga and Kamrin, 2022), and has also recently been applied to other geophysical systems (e.g., Damsgaard et al., 2020; Zhang et al., 2022)

In the nonlocal granular fluidity rheology, the effective coefficient of friction depends on the granular fluidity, $g'$, which is a measure of how easily the material can flow for a given stress:

$$\mu \equiv \frac{\dot{\epsilon}_e}{g'}. \tag{13}$$

The granular fluidity depends on local and distant stresses through the differential relation

$$\nabla^2 g' = \frac{1}{\xi^2}\left(g' - g'_{\text{loc}}\right), \tag{14}$$

where $\xi$ is the cooperativity length and $g'_{\text{loc}}$ is the local granular fluidity. The local granular fluidity is based on experiments that suggest that granular materials behave like Bingham fluids (solid at low stresses and viscous at high stresses):

$$g'_{\text{loc}} = \begin{cases} \sqrt{\dfrac{\tilde{P}}{\rho d^2}} \dfrac{(\mu - \mu_s)}{\mu b} & \text{if } \mu > \mu_s \\ 0 & \text{if } \mu \leq \mu_s, \end{cases} \tag{15}$$

where $b$ is a dimensionless constant and $\mu_s$ is the static yield coefficient. The Laplacian term in Equation (14) spreads out the fluidity into regions where $\mu < \mu_s$ (Kamrin and Koval, 2012) and allows for deformation in regions of low stress. The distance over which the fluidity spreads out is determined by the cooperativity length, which scales with grain size and diverges at the yield point (Bocquet et al., 2009; Kamrin and Henann, 2015):

$$\xi = \frac{Ad}{\sqrt{|\mu - \mu_s|}}, \tag{16}$$

where $A$ is a dimensionless constant.

Substituting Equation (13) into Equation (12) yields

$$\frac{\partial}{\partial x}\left[\frac{H^2}{g' + \dot{\epsilon}_{xx} + \dot{\epsilon}_{yy}}(2\dot{\epsilon}_{xx} + \dot{\epsilon}_{yy})\right] + H\frac{\partial}{\partial y}\left[\frac{H}{g' + \dot{\epsilon}_{xx} + \dot{\epsilon}_{yy}}\dot{\epsilon}_{xy}\right] = 2H\frac{\partial H}{\partial x}$$
$$\frac{\partial}{\partial y}\left[\frac{H^2}{g' + \dot{\epsilon}_{xx} + \dot{\epsilon}_{yy}}(2\dot{\epsilon}_{yy} + \dot{\epsilon}_{xx})\right] + H\frac{\partial}{\partial x}\left[\frac{H}{g' + \dot{\epsilon}_{xx} + \dot{\epsilon}_{yy}}\dot{\epsilon}_{xy}\right] = 2H\frac{\partial H}{\partial y}. \tag{17}$$

Equation (17), along with the equations for $g'$ (Equations 13–16), is analogous to the shallow shelf approximation. We therefore suggest referring to Equation (17) as the nonlocal shallow mélange approximation (NSMA).





## 2.2 Width-integrated flow equations and boundary conditions

To reduce Equation (17) to a quasi-one-dimensional flow model we adopt an approach from glacier flow modeling in which extension-dominated dynamics are used to characterize the longitudinal stresses and shear-dominated dynamics are used to characterize the shear stresses (Pegler, 2016). This approach allows for width-integration of the flow equations and, importantly, asymptotes to the correct dynamics in extension- and shear-dominated regimes. Essentially, we assume that (i) flow is in the $x$-direction and variations in width are small (i.e., $|dW/dx| \ll 1$; Pegler, 2016), so that $v \approx 0$ and $\dot{\epsilon}_{yy} \approx 0$, (ii) the ice mélange

thickness and longitudinal strain rates are uniform across the width of the fjord, and (iii) the granular fluidity in the longitudinal stress term ($R_{xx}$) is only a function of $\dot{\epsilon}_{xx}$ while the granular fluidity in the shear stress term ($R_{xy}$) is only a function of $\dot{\epsilon}_{xy}$.

Under these assumptions, integrating the $x$-component of Equation (17) across the fjord and dividing by the width $W$ yields

$$\frac{\partial}{\partial x}\left[\frac{H^2}{g^x + \dot{\epsilon}_{xx}}\dot{\epsilon}_{xx}\right] - \frac{H^2}{W}\mu_w\left(\frac{\dot{\epsilon}_{xy}}{\dot{\epsilon}_e}\right)_{y=0} = H\frac{\partial H}{\partial x}, \tag{18}$$

where $g^x$ is used to indicate that the granular fluidity in the longitudinal stress term depends solely on $\dot{\epsilon}_{xx}$, $\mu_w$ is the value of $\mu$

along the fjord walls, $y$ is taken to be the distance from the near wall of the fjord, and due to symmetry the shear strain rates at $y = 0$ and $y = W$ have the same magnitude but opposite sign. Due to our assumptions, the $y$-component of Equation (17) does not affect flow in the $x$-direction and can be ignored. The first and second terms in Equation (18) characterize extension- and shear-dominated dynamics, respectively.

For shear-dominated flow, $(\dot{\epsilon}_{xy}/\dot{\epsilon}_e)|_{y=0} = \text{sgn}(\dot{\epsilon}_{xy})|_{y=0} = \text{sgn}(U)$, where $U$ is the depth- and width-averaged velocity.

Thus, combining and rearranging Equation (18) gives the one-dimensional stress balance equation:

$$\frac{\partial}{\partial x}\left[\frac{H^2}{g^x + (\partial U/\partial x)}\frac{\partial U}{\partial x}\right] = H\frac{\partial H}{\partial x} + \frac{H^2}{W}\mu_w\,\text{sgn}(U). \tag{19}$$

Equation (19) is the key dynamical equation that is used to determine the ice mélange velocity along the length of the fjord. We define $x = 0$ as being the upstream end of the ice mélange. At this boundary, material flows into the domain at a rate determined by the iceberg calving flux. Conservation of mass dictates that the velocity there is given by

$$U_0 = U_c\frac{H_t}{H_0}, \tag{20}$$

where subscript 0 refers to values at $x = 0$, $U_c$ is the calving rate, and $H_t$ is the terminus thickness. We define the downstream end of the ice mélange ($x = L$) as being the point where the ice thins to the grain scale, $d$. At thicknesses less than grain scale, the nonlocal granular fluidity rheology no longer applies. In order to prevent divergence for thicknesses less than the grain scale, we therefore require that the velocity gradient there is

$$\left.\frac{\partial U}{\partial x}\right|_{x=L} = 0. \tag{21}$$

This downstream boundary condition is similar to regularizations used in sea ice models to prevent ice floes with free boundaries from spreading under their own weight (Hibler, 2001; Leppäranta, 2012).

The granular fluidity $g^x$ is described by a simplified version of Equation (14) in which $g'$ and $g'_{\text{loc}}$ are replaced with $g^x$ and $g^x_{\text{loc}}$, $\nabla^2 g^x = \partial^2 g^x/\partial x^2$, and $\dot{\epsilon}_e = |\dot{\epsilon}_{xx}| + \delta_{\dot{\epsilon}}$. $\delta_{\dot{\epsilon}}$ is a strain rate parameter that is used to regularize the granular fluidity





equations in order to improve stability and efficiency. Other regularization schemes are possible (Chauchat and Médale, 2014); however, we have had success with this simple regularization scheme and therefore leave investigation of other schemes for future work. For boundary conditions we set $\partial g^x / \partial x = 0$ at $x = 0, L$ following the recommendation of Henann and Kamrin (2013).

The value of $\mu_w$ is related to the width-averaged velocity relative to the fjord walls, which is given by $U + (U_t - U_c)$, where $U_t$ is the glacier terminus velocity and $U_t - U_c$ is the rate of glacier terminus migration. In other words, the fjord walls move backward in our coordinate system, which is defined relative to the glacier terminus, at a rate given by $U_t - U_c$. For shear-dominated flow, the effective coefficient of friction varies linearly across the fjord, such that

$$\mu = \mu_w \left( 1 - \frac{2y}{W} \right) \tag{22}$$

for $0 \leq y \leq W/2$. Using Equation (22), the local granular fluidity and cooperativity length can be readily calculated as functions of position for a given value of $\mu_w$. The results are then inserted into the granular fluidity differential equation (Equation 14), except that $g'$ and $g'_{\text{loc}}$ are replaced with $g^y$ and $g^y_{\text{loc}}$ (to emphasize that the granular fluidity for shear-dominated flow depends only on $\dot{\epsilon}_{xy}$) and $\nabla^2 g^y = \partial^2 g^y / \partial y^2$. As before, we set $\partial g^y / \partial y = 0$ at both boundaries. The granular fluidity equation is then solved to determine $g^y(y, \mu_w)$. If the flow is in the positive $x$-direction then $\dot{\epsilon}_e = (\partial U / \partial y)/2$ and Equation (13) can be rewritten as

$$\frac{\partial u}{\partial y} = 2 \mu g^y = 2 \mu_w \left( 1 - \frac{2y}{W} \right) g^y(y, \mu_w). \tag{23}$$

The average velocity in the transect is found by integrating Equation (23), which must equal the velocity in the bedrock reference frame:

$$U + U_t - U_c = \frac{2}{W} \int\limits_0^{W/2} \int\limits_0^y 2 \mu_w \left( 1 - \frac{2y'}{W} \right) g^y(y', \mu_w) \, dy' \, dy. \tag{24}$$

Finally, the ice mélange geometry changes in response to melting, flow divergence, and dispersal of icebergs at $x = L$. The surface evolves according to the depth- and width-integrated mass continuity equation (van der Veen, 2013), in which

$$\frac{\partial H}{\partial t} = \dot{B} - \frac{1}{W} \frac{\partial}{\partial x} (UHW), \tag{25}$$

where $\dot{B}$ is the surface plus basal mass balance rate, and the length evolves so as to ensure that the thickness at the end of the ice mélange is always equal to the characteristic iceberg size.

## 2.3 Numerical implementation and stability considerations

The quasi-one-dimensional ice mélange flow model that we have developed depends on five variables: $U$, $g^x$, $\mu_w$, $H$, and $L$. We determine these variables by simultaneously solving the width-integrated NSMA, granular fluidity, transverse velocity, and mass continuity equations (Equations 19, 14, 24, and 25), while also requiring that $H_L = d$. We use finite differences with a stretched coordinate system and a staggered grid for velocity and thickness. The mass continuity equation uses an implicit



**Table 1.** Description of model variables.

| Variable | Description |
| --- | --- |
| $\rho, \rho_w$ | densities of ice and water |
| $x_i = \langle x, y, z \rangle$ | position vector |
| $g, g_{\mathrm{eff}}, t$ | gravitational acceleration, effective gravity, and time |
| $u_i = \langle u, v, w \rangle$ | velocity vector |
| $\dot{\epsilon}_{ij}, \dot{\epsilon}_e$ | strain rate tensor and effective strain rate |
| $\sigma_{ij}, \sigma'_{ij}, R_{ij}$ | stress, deviatoric stress, and tectonic stress tensors |
| $\delta_{ij}$ | kronecker delta |
| $\tilde{p}$ | granular static pressure |
| $P, \tilde{P}$ | depth-averaged pressure and granular static pressure |
| $U, H, W, L$ | ice mélange velocity, thickness, width, and length |
| $H_t, U_t, U_c$ | depth- and width-averaged (glacier) terminus thickness, terminus velocity, and calving rate |
| $U_0, H_0$ | velocity and thickness at $x = 0$ |
| $F$ | buttressing force |
| $\dot{B}$ | surface plus basal mass balance rate |
| $\mu, \mu_w$ | effective coefficient of friction within the ice mélange and along the fjord walls |
| $\mu_s$ | static yield coefficient |
| $g', g'_{\mathrm{loc}}$ | granular fluidity and local granular fluidity |
| $g^x, g^y$ | granular fluidity for extension-dominated and shear-dominated flow |
| $\xi$ | cooperativity length |
| $d$ | characteristic iceberg diameter |
| $b, A$ | dimensionless parameters |
| $\delta\dot{\epsilon}$ | finite strain rate parameter |
| $\chi, \tau, \dot{\epsilon}_\chi$ | longitudinal position, time, and effective strain rate in the stretched coordinate system |

time stepping scheme and an upwind scheme for discretization. Our numerical scheme, which closely mimics that of Schoof

195  (2007), is described in detail in the Appendix.

The width-integrated NSMA is more computationally expensive than the analogous width-integrated SSA approximation for two reasons. First, the nonlocal granular fluidity rheology introduces additional nonlinear differential equations that must be solved as part of the iteration procedure, essentially doubling the number of unknowns. Second, because ice mélange tends to be considerably thinner than its parent glacier, ice mélange velocities must be several times higher than glacier terminus

200  velocities in order to balance the ice flux into the fjord. This latter effect becomes particularly critical if ice mélange thins to close to its characteristic iceberg size.



For example, although we are using an implicit scheme, we find that the CFL condition (Courant-Fredrichs-Lewy; $\Delta t \leq C_{\max} \Delta x / U$) is a useful metric for determining appropriate time steps that maintain numerical stability. From our experience, $C_{\max} = 1$ provides good stability across a range of parameter choices and model states, although this is not a strict requirement. At $x_0$, the ice mélange velocity is $U_0 = U_c H_t / H_0$ (Equation 20). Thus the CFL condition at $x_0$ can be expressed as

$$\Delta t \leq \frac{\Delta x H_0}{U_c H_t}. \tag{26}$$

For thick ice mélange ($H_0 \approx H_t$), with a calving rate of 6000 m a$^{-1}$, a terminus thickness of $H_t = 600$ m, and a grid spacing of 500 m, $\Delta t < 0.09$ a. However if the ice mélange approaches the characteristic iceberg size, for example $H_0 \approx d = 25$ m, then $\Delta t < 4 \times 10^{-3}$ a (assuming a similar grid spacing). In reality, higher velocities may occur farther down fjord, necessitating shorter time steps. Since our model uses a moving grid and the ice mélange thickness and length may vary significantly over seasonal time scales, we recommend using short time steps or an adaptive time step in prognostic simulations.

### 2.4 Ice mélange buttressing force

Although we do not model glacier flow in this paper, we do assess the impact of model parameters, glacier fluxes, melt rates, and fjord geometry on the buttressing force that ice mélange exerts on glacier termini, which is given by $(-HW\sigma_{xx})|_{x_0}$. The force imposed on a glacier terminus (per unit width) due to the presence of ice mélange is therefore

$$F/W = \left(-H R_{xx} + H\tilde{P}\right)_{x_0}. \tag{27}$$

Substituting in the nonlocal granular fluidity rheology yields

$$F/W = \left(-\frac{2H\tilde{P}\dot{\epsilon}_{xx}}{g^x + \dot{\epsilon}_{xx}} + H\tilde{P}\right)_{x_0}. \tag{28}$$

In the limit that $\partial U / \partial x \to 0$, $F/W$ scales with the thickness squared, $H_0^2$.

## 3 Model results

### 3.1 Steady-state profiles

We begin exploring the model behavior by investigating the impact of model parameters and forcings on steady-state profiles. The model is capable of producing two types of steady state solutions: one in which material is continuously flowing through the ice mélange domain and the geometry is steady in the bedrock reference frame, and one in which no material enters or leaves the ice mélange and the geometry is steady in a reference frame that moves down fjord with the glacier terminus and the velocity is constant ($\partial U / \partial x = 0$). We refer to these two states as the "steady-state" and "quasi-static" regimes. We focus primarily on the steady-state regime as the quasi-static regime has already been analyzed in some detail in Amundson and Burton (2018).



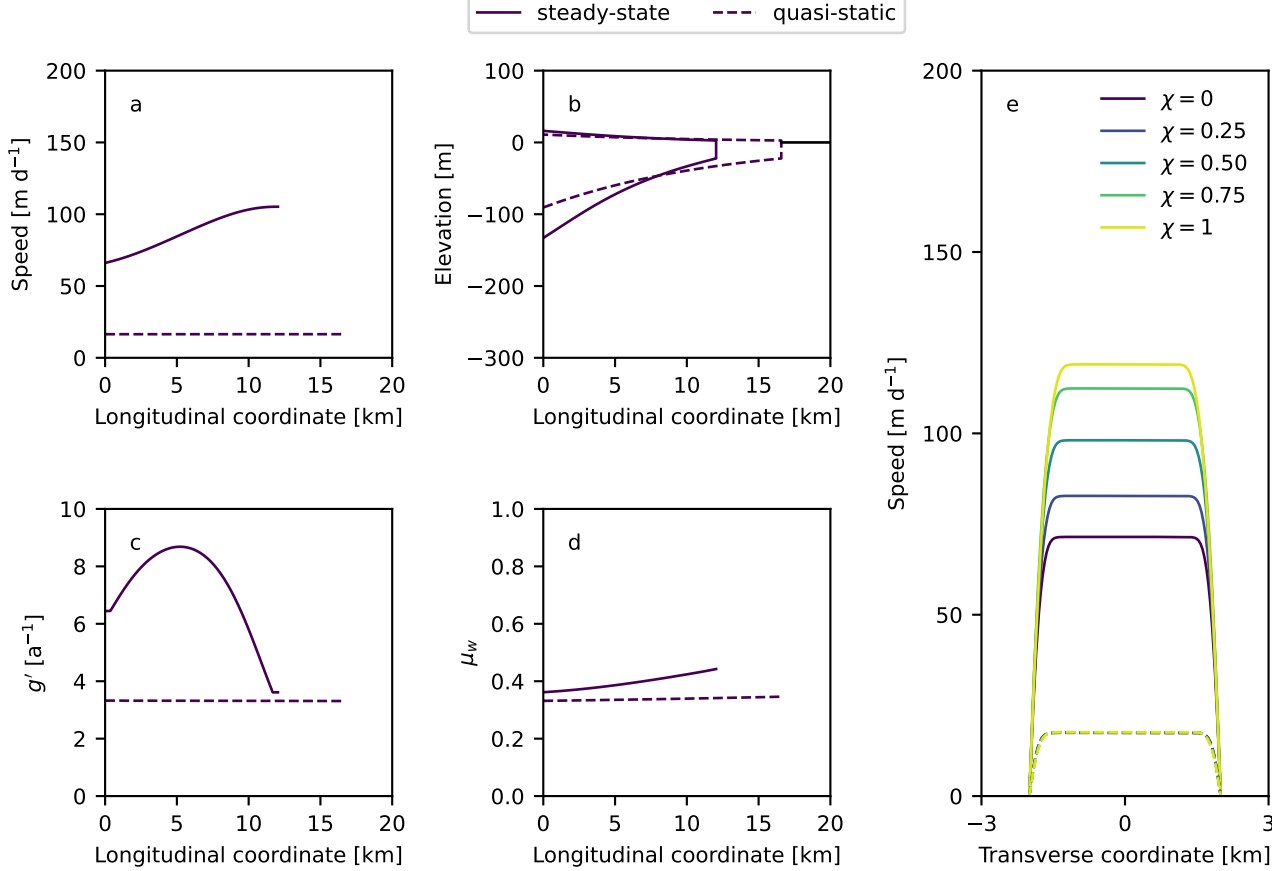

**Figure 1.** Steady-state (solid curves) and quasi-static (dashed curves) profiles. (a)–(d) Longitudinal profiles of velocity, thickness, granular fluidity, and limit of internal friction along the fjord walls. (e) Transverse velocity profiles at various fractions $\chi$ of the distance along the ice mélange. For the steady-state simulation, $U_t = U_c = 6000$ m a$^{-1}$, $H_t = 600$ m, $W = 4000$ m, $\dot{B} = -0.6$ m d$^{-1}$, $d = 25$ m, $\mu_s = 0.3$, $A = 0.5$, and $b = 1 \times 10^4$. Longitudinal coordinates are relative to the glacier terminus.

To produce steady-state profiles, we set the terminus velocity and calving rate to be constant and equal to each other and

we set the surface mass balance rate equal to a constant. We then run prognostic simulations until the ice mélange length and thickness are no longer changing with time ($dL/dt = 0$ and $\partial H/\partial t = 0$). The approach that we adopt here differs from that of Amundson and Burton (2018), where we derived an expression for steady-state profiles in the quasi-static limit in several important ways: (i) we do not set the calving and melt rates equal to 0, (ii) we do not require $\mu_w$ to be constant but instead solve for it, (iii) we allow for variable width, and (iv) the ice mélange length is not specified a priori but rather is determined

by the balance of the inflow and melt rates. We then turn off calving and melting and allow the ice mélange to evolve to a new




steady-state in order to demonstrate the changes in flow and geometry that occur during the transition from the steady-state regime to the quasi-static regime.

Example steady-state and quasi-static profiles are shown in Figure 1. Velocities increase in the down fjord direction in the steady-state scenario which, when combined with surface and basal melting, leads to a relatively large thickness gradient. The

extensional flow is also associated with an increase in $\mu_w$ in the down fjord direction. Once calving and melting are turned off the velocities drop because there is no longer a flux of material into the ice mélange and the icebergs are simply pushed at the rate of glacier terminus advance. Consequently the shear stresses also decrease, which is reflected in a decrease in $\mu_w$. The reduction in shear stresses allows the ice mélange to thin and spread outward. When the quasi-static limit is reached the ice mélange has a roughly exponential thickness profile and $\mu_w$ is spatially constant. In Amundson and Burton (2018) we assumed

that $\mu_w$ is a constant in the quasi-static limit and showed that this leads to a roughly exponential thickness profile; here we see it arise naturally through the momentum and mass continuity equations in a manner that is consistent with our prior assumptions.

### 3.1.1  Sensitivity to model parameters

The nonlocal granular fluidity rheology depends on several parameters that must be specified: the characteristic iceberg size $d$, dimensionless constants $b$ and $A$ (described below), and the static yield coefficient $\mu_s$. For default values we have selected

$d = 25\,\text{m}$, $b = 1 \times 10^4$, $A = 0.5$, and $\mu_s = 0.3$, which produces thickness and velocity profiles that are roughly consistent with observations from Sermeq Kujalleq (Jakobshavn Isbræ), Helheim Glacier, and Kangerdlugssuaq Glacier (e.g., Foga et al., 2014; Amundson and Burton, 2018; Xie et al., 2019). Here, we provide some context for our selection of default values and explore how adjusting these parameters affects the model behavior (refer to Figure 2 throughout this section).

- The characteristic iceberg size influences the local granular fluidity, the cooperativity length (see Equations 15 and 16),
and the ice mélange extent (since the end of the domain is defined as being where $H = d$). Ice mélange is a highly heterogeneous material, with iceberg dimensions ranging from meters to hundreds of meters. Several studies indicate that iceberg area (in map view) follows power-law size distributions, $p(a) \propto a^{-\alpha}$, with $\alpha$ ranging from 2.1–3.4 (e.g., Enderlin et al., 2016; Sulak et al., 2017; Kirkham et al., 2017; Kaluzienski et al., 2023). Power law distributions require a minimum size threshold. Using a minimum area of 10 m$^2$ gives median and mean iceberg areas of about 13–18 m$^2$
and 17–110 m$^2$ (see Equations 6 and 8 in Kaluzienski et al., 2023), resulting in a characteristic diameter on the order of 4–10 m. It is unclear, however, how iceberg heterogeneity affects ice mélange flow or if there is a controlling iceberg size. Nonetheless, we find that decreasing the iceberg size allows the ice mélange to thin and advance.

- Dimensionless constant $b$ is given by the ratio of the range of effective friction coefficients to the inertial number (see Kamrin and Henann, 2015), which is itself a function of grain size, characteristic strain rate, and pressure. These values
are poorly constrained for ice mélange at present. Using typical values, we find that $b$ is likely in the range of $10^4$–$10^6$. $b$ only affects the local granular fluidity (Equation 15), and as such its impact on model behavior is more transparent than that of iceberg size. Increasing $b$ makes the ice mélange more stiff and extensive.



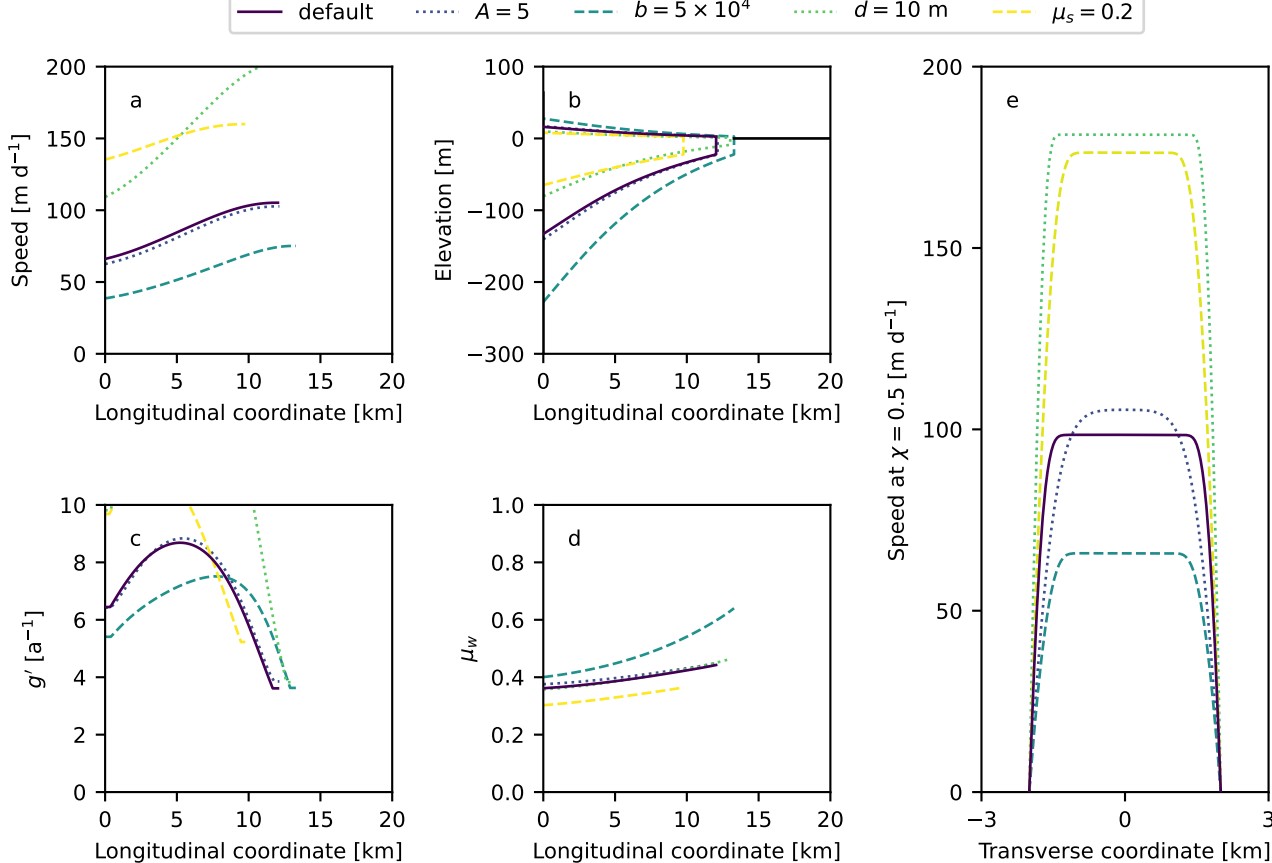

**Figure 2.** Steady-state profiles for various model parameter choices. For all simulations, $U_t = U_c = 6000$ m a$^{-1}$, $\dot{B} = -0.6$ m d$^{-1}$, $H_t = 600$ m, and $W = 4000$ m. The solid curves represent the results produced using the default values ($A = 0.5$, $b = 1 \times 10^4$, $d = 25$ m, and $\mu_s = 0.3$). For the other curves, we adjusted one model parameter, as indicated in the legend, but kept all other parameters set to their default values. Longitudinal coordinates are relative to the glacier terminus.

– Dimensionless constant $A$ affects the cooperativity length and is thought to be of order one; fitting to laboratory experiments and discrete element simulations suggests that $A$ equals 0.5 for glass beads and 0.9 for stiff disks (Henann and Kamrin, 2013; Kamrin and Koval, 2014). For our simulations, using values of $A = 0.5$ gives cooperativity lengths of a few kilometers in the longitudinal direction. Changing $A$ does not have much impact on our results other than changing the curvature of the transverse velocity profiles.

– Lastly, the static yield coefficient determines the stress at which the ice mélange will begin to flow. Reducing the yield coefficient causes the ice mélange to deform more easily and become thinner and shorter.






Determining appropriate model parameters that are able to describe ice mélange flow across a range of forcings and fjord geometries remains a major task. The default parameters that we have selected produce ice mélange geometries and velocity profiles that appear to be roughly consistent with field observations. Adjusting any of the parameters appreciably from our default parameters will likely require modifying one or more additional parameters in order to produce profiles that are not too thin or too thick. For example, we can also produce similar profiles if we reduce the static yield coefficient but only if we

increase $b$ appropriately.

### 3.1.2 Sensitivity to external forcings and fjord geometry

The modeled ice mélange geometry and flow depend on iceberg calving fluxes, surface and basal melt rates, and fjord geometry. We address each of these in turn.

    To investigate the impact of calving fluxes on ice mélange flow and geometry, we consider three scenarios in which the

glacier velocity and calving rate scale with the fjord width and glacier terminus thickness. The ice mélange becomes more extensive as the fluxes increase (Figure 3), implying that ice mélange produced by highly active glaciers is more likely to exert high resistive stresses against the glacier termini and to persist year round. One way of estimating the minimum force from ice mélange that will affect calving is by considering the torques acting on a full-glacier-thickness iceberg that has detached from the glacier but not yet capsized. The buoyant torque acting on the iceberg scales with $H_t^3$ (Section 3.2 in Burton et al., 2012)

and therefore the ice mélange buttressing force that would prevent an iceberg from capsizing scales with $H_t^2$ (see Equation 1 in Amundson et al., 2010). In these simulations we varied the terminus thickness $H_t$ from 600–800 m and varied the calving rate from 6000–8000 m a$^{-1}$. The force that would be required to prevent large icebergs from capsizing increased by $\sim 77\%$, yet the buttressing force increased by about 100% because the ice mélange thickness increased by over 40%. Although the imposed calving rates are ad hoc, these results suggest that highly productive glaciers are more likely to be affected by ice mélange

buttressing because as $H_t$ increases, the buttressing force from the resulting ice mélange increases more rapidly than the force required to prevent icebergs from capsizing.

    The ocean affects the modeled ice mélange by melting it. Iceberg melt rates in fjords can range from 0.1–0.8 m d$^{-1}$ (Enderlin et al., 2016) and icebergs are particularly important sources of freshwater in winter (Moon et al., 2017). We find that ice mélange extent is sensitive to melt due to its indirect effect on lateral shear stresses (Figure 4) and that the buttressing force depends on

the melt rate through an inverse power law relationship with an exponent of about -3.

    Fjord width also has important impacts on ice mélange extent and buttressing force. Increasing the fjord width reduces the ability of shear stresses to build an ice mélange wedge, and thus the ice mélange thins and sheds icebergs. Consequently the buttressing force decays roughly exponentially with fjord width (Figure 5 a–b), as also observed in the analysis of quasi-static flow (Amundson and Burton, 2018; Burton et al., 2018). The width gradient has similar effects on the buttressing force.

Converging walls ($dW/dx < 0$) create extra flow resistance that allows for the development of a thicker ice mélange wedge. The buttressing force also decays roughly exponentially with the width gradient (Figure 5c–d).



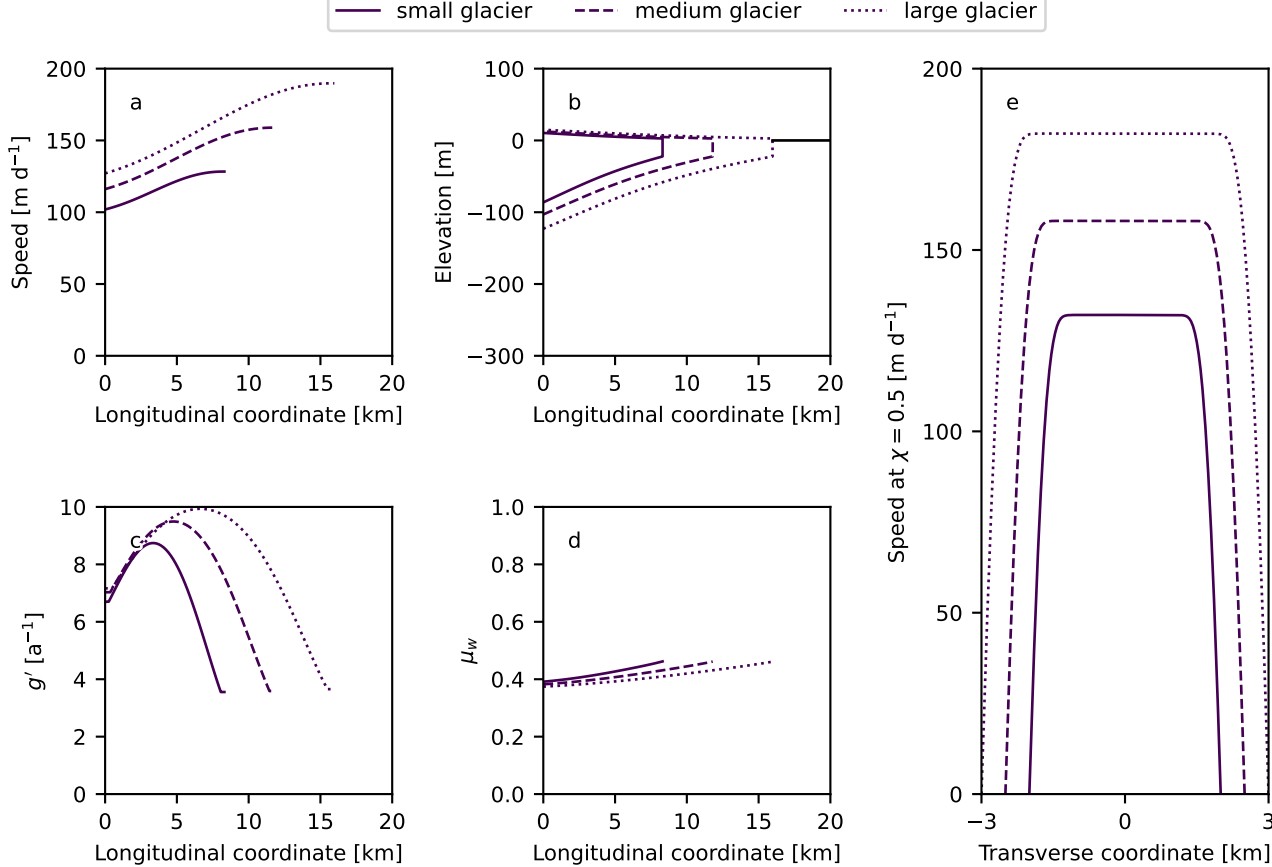

**Figure 3.** Steady-state profiles for various size glaciers as follows: (solid lines) $U_t = U_c = 6000$ m a$^{-1}$, $H_t = 600$ m, and $W = 4000$ m; (dashed lines) $U_t = U_c = 7000$ m a$^{-1}$, $H_t = 700$ m, and $W = 5000$ m; (dotted lines) $U_t = U_c = 8000$ m a$^{-1}$, $H_t = 800$ m, and $W = 6000$ m. (a)–(d) Longitudinal profiles of velocity, thickness, granular fluidity, and coefficient of friction along the fjord walls. (e) Transverse velocity profiles at the ice mélange midpoint. For all simulations, $\dot{B} = -0.8$ m d$^{-1}$, $d = 25$ m, $\mu_s = 0.3$, $A = 0.5$, and $b = 1 \times 10^4$. Longitudinal coordinates are relative to the glacier terminus.

## 3.2 Transient simulations

The ice mélange buttressing force is clearly sensitive to changes in ice mélange thickness. From field and remote sensing observations we expect ice mélange to be weakest in summer, when melt rates and calving activity are highest (e.g., Joughin

et al., 2020). To investigate the implications of these fluctuations, we impose seasonal variations in melting and calving rates with amplitudes of 0.2 m d$^{-1}$ and 600 m a$^{-1}$, respectively.

We find that the buttressing force decreases as the melt rate increases (Figure 6a–b), as might be expected during the summer months. However, there is a lag of 2 months between the highest melt rates and the weakest ice mélange. The lag is smallest





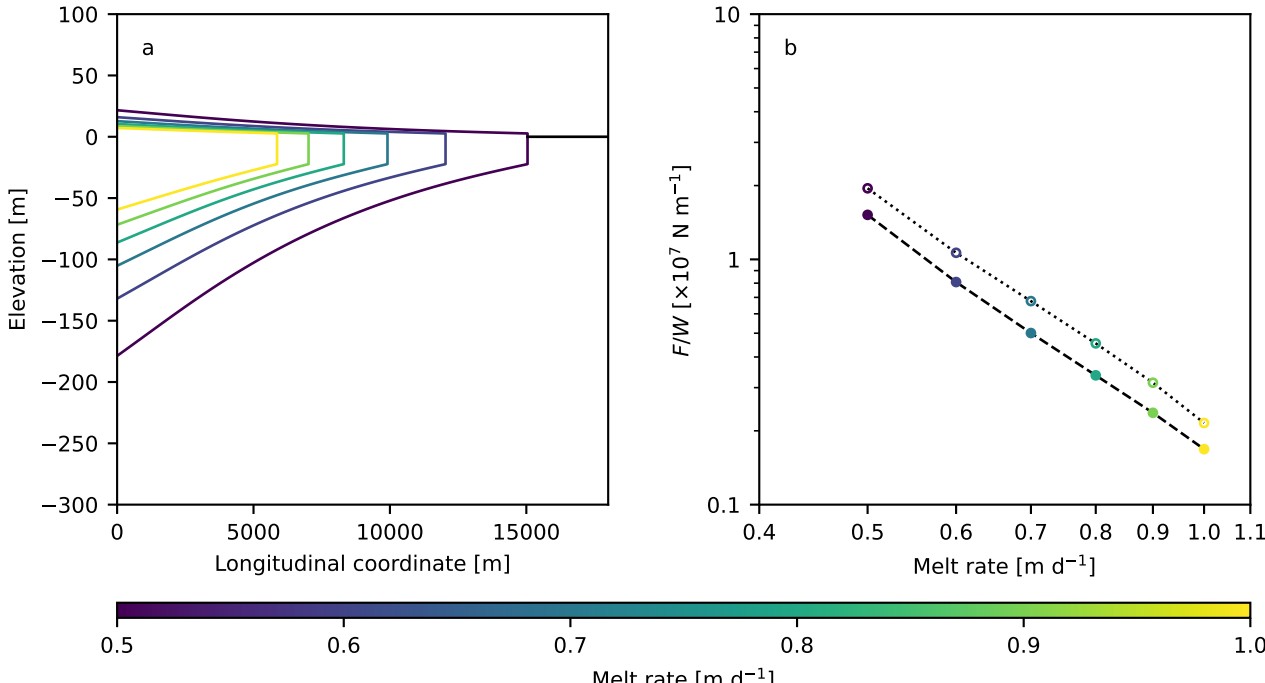

**Figure 4.** Effect of melt rates on (a) steady-state thickness profiles and (b) the ice mélange buttressing force per unit width. The dotted curve represents force estimates computed by neglecting longitudinal strain rates, whereas the dashed curve includes the effect of longitudinal strain rates. For all simulations, $U_t = U_c = 6000$ m a$^{-1}$, $H_t = 600$ m, $W = 4000$ m, $d = 25$ m, $\mu_s = 0.3$, $A = 0.5$, and $b = 1 \times 10^4$.

for ice mélange experiencing higher melt rates because smaller ice mélange will respond more rapidly to external forcings.

There is also less variability in the buttressing force for smaller ice mélange.

Iceberg calving also varies seasonally and tends to be highest in the summer. The model, which assumes the ice mélange remains well packed year round, predicts that it will thicken and grow in response to the addition of new material. As with melting, there is a lag of 2 months between variations in calving rates and the force exerted on the glacier termini, and the amplitude of the variations in force also decrease with ice mélange extent (Figure 6c–d). Thus, melt and calving, which both

vary seasonally, have opposite effects on the model behavior.

Following observations that suggest that iceberg calving is affected by the ice mélange buttressing force, we use an ad hoc linear relationship between calving and the buttressing force to begin investigating their coupled impacts on ice mélange. We suppose that

$$U_c = 2U_{c,ss} - \frac{U_{c,ss}}{F_{ss}} F, \tag{29}$$

where $U_{c,ss}$ and $F_{ss}$ are the steady-state calving rate and buttressing force for a given set of model parameters. An imposed variation in melt rates causes $F$ to vary, which is coupled to the calving rate via a negative feedback loop. This coupling reduces





**Figure 5.** Effect of fjord geometry on steady-state thickness profiles and the buttressing force per unit width. In (a) and (b) the width was varied while the gradient in width was held constant. In (c) and (d) the width at the glacier terminus was fixed at 4000 m and the gradient in width was varied. Positive (negative) values of $dW/dx$ correspond to fjord walls that are diverging (converging). The dotted curves represent force estimates computed by neglecting longitudinal strain rates, whereas the dashed curves includes the effect of longitudinal strain rates. For all simulations, $U_t = U_c = 6000$ m a$^{-1}$, $H_t = 600$ m, $d = 25$ m, $\mu_s = 0.3$, $A = 0.5$, and $b = 1 \times 10^4$.



the lag time between the melt rate and the buttressing force to about 0.1 a and, as a result, the calving rate is high when melt rates are also high (Figure 6).

### 3.3 Buttressing forces in the steady-state and quasi-static regimes

The ice mélange buttressing force depends on the thickness, tectonic stress $R_{xx}$, and granular static pressure $\tilde{P}$ at the glacier-ice mélange boundary (Equation 28). In the quasi-static limit the velocity gradient is zero and therefore the buttressing force scales with $H_0^2$. However, we find that when calving and melting are nonzero the flow is extensional, which causes the buttressing force to be less than would be expected if buttressing force estimates were based solely on ice thickness (Figs. 4 and 5).

We find that, for parallel-sided fjords, the buttressing force in the steady-state regime also scales with $H_0^2$ despite the com-
plexity introduced by nonzero strain rates (Fig. 7). During our transient simulations the buttressing force circles around the initial steady-state solutions as the flow becomes more/less extensional. We never observe compressional flow in our simulations, and field and remote sensing observations indicate that compressional flow only occurs during and in the immediate aftermath of iceberg calving events (Peters et al., 2015; Amundson and Burton, 2018; Cassotto et al., 2021). Thus, observations of ice mélange thickness from satellite data, along with the quasi-static approximation of Equation (28), can be used to provide
an upperbound on the ice mélange buttressing force.

### 4 Conclusions

We have developed a depth-averaged continuum model of ice mélange flow, which we refer to as the nonlocal shallow mélange approximation, that is based on recent advances in our understanding of granular materials and that is suitable for long time-scale glacier simulations. Consistent with other granular flows, the model exhibits viscous flow where the stresses are far from
the yield point and plug flow where the stresses approach the yield point.

The model contains four parameters (the iceberg size, two dimensionless constants, and the static yield coefficient) that must be tuned. We have selected a set of parameters that produce velocity and thickness profiles that are roughly consistent with remote sensing observations from Greenland (Foga et al., 2014; Amundson and Burton, 2018; Xie et al., 2019). Ultimately, the profiles depend on the ice mélange stiffness; stiff ice mélange does not spread very easily and tends to result in thick, extensive
ice coverage. Each of the four model parameters can affect the overall fluidity; thus, other parameter combinations may also produce suitable model results. Determining the best parameter values that work across a range of forcing and fjord geometries remains a major task for laboratory experiments and field observations.

We assume that the ice mélange is well packed and homogeneous, and we do not account for cohesion. The model is likely to perform best for winter ice mélange and for systems where ice mélange persists year round since the flow approximation
is not applicable for granular materials far from the well-packed limit. The impacts of iceberg heterogeneity and cohesion on ice mélange flow require further investigation. We suggest that both could potentially be incorporated into our modeling framework through modification of the model parameters, which are currently treated as constants, and/or by tuning the model parameters with field observations, laboratory experiments, and discrete element simulations. Future work should also attempt





**Figure 6.** Ice mélange response to temporally varying melt rates and calving rates. (a) Sinusoidal variations in melt rate (dashed curve) and buttressing force (solid curves). (b) Hysteresis curves between buttressing force and melt rate for different baseline melt rates (and therefore ice mélange sizes). (c) Sinusoidal variations in calving rates (dashed lines) and buttressing force (solid lines). (d) Hysteresis curves between buttressing force and calving rate for different baseline melt rates (same as in panel b). The starting points of the hysteresis curves in (b) and (d) are indicated by dots. The black curves in (a) and (b) correspond to a simulation in which the calving rate depends linearly on the buttressing force. For all simulations, $d = 25$ m, $\mu_s = 0.3$, $A = 0.5$, and $b = 1 \times 10^4$.



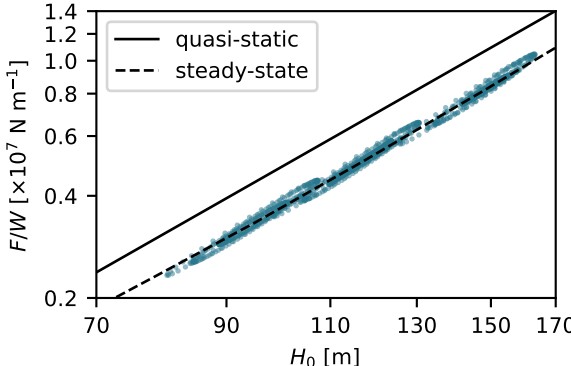

**Figure 7.** Relationship between ice mélange buttressing force and thickness at $x_0$. The solid black line represents the quasi-static regime, the dashed black line represents the results from steady-state solutions shown in Figure 4, and the colored dots represent the solutions for all of the transient simulations shown in Figure 6.

to quantify the degree to which the quasi-one-dimensional model can replicate the behavior of ice mélange in fjords with complex geometry.

Ultimately, we find that the nonlocal shallow mélange approximation produces realistic (albeit smooth) thickness and velocity profiles and evolves in response to glaciological, atmospheric, and oceanographic forcing. Ice fluxes, melt rates, and fjord geometry strongly affect the model geometry and ice mélange buttressing forces. Addition of new material into the ice mélange via iceberg calving makes it longer, thicker, and more resistive, whereas removal of material through surface and basal melting does the opposite. Thus the model may be capable of explaining temporal variations in buttressing forces and why ice mélange appears to have larger impacts in some glacier-fjord systems than others.

*Code availability.* The model code (glaciome1D) and files used to produce the figures in this manuscript are available at https://github.com/jmamundson/glaciome1D. The code is written in Python and uses standard Python libraries.

## Appendix A: Coordinate stretching

We use a coordinate system that moves with the glacier terminus and, following Schoof (2007), we introduce a coordinate stretching to deal with the moving boundary at the end of the ice mélange ($x = L$):

$$\chi = \frac{x}{L}, \tag{A1}$$

which maps $0 \leq x \leq L$ to $0 \leq \chi \leq 1$. According to the chain rule,

$$\frac{\partial}{\partial x} = \frac{\partial \chi}{\partial x} \frac{\partial}{\partial \chi} = \frac{1}{L} \frac{\partial}{\partial \chi}. \tag{A2}$$





The coordinate stretching also necessitates a transformation of time derivatives. The material derivative is

$$\frac{D}{Dt} = \frac{dx}{dt}\frac{\partial}{\partial x} + \frac{\partial}{\partial t}. \tag{A3}$$

The grid points move with velocity

$$\frac{dx}{dt} = \chi\frac{dL}{dt}. \tag{A4}$$

The material derivative of a quantity that is moving with the grid is the same as the partial derivative of that same quantity
in the grid's reference frame. As in Schoof (2007), we therefore let $t = \tau$ to distinguish between partial derivatives when $x$
and $\chi$ are held constant, respectively, which allows us to replace $D/Dt$ with $\partial/\partial\tau$. Thus, after rearranging Equation (A3) and
inserting Equations (A2) and (A4), we arrive at

$$\frac{\partial}{\partial t} = \frac{\partial}{\partial \tau} - \frac{\chi}{L}\frac{dL}{d\tau}\frac{\partial}{\partial \chi}. \tag{A5}$$

The coordinate transformations are then applied to the stress balance, granular fluidity, and mass continuity equations (Equa-
tions 19, 14, and 25), yielding

$$\frac{1}{L}\frac{\partial}{\partial \chi}\left(\frac{1}{g^x + (\partial U/\partial\chi)/L}H^2\frac{1}{L}\frac{\partial U}{\partial \chi}\right) = \frac{H}{L}\frac{\partial H}{\partial \chi} + \frac{H^2}{W}\mu_w\,\mathrm{sgn}(U)$$
$$\frac{1}{L^2}\frac{\partial^2 g^x}{\partial \chi^2} = \frac{1}{\xi^2}\left(g^x - g^x_{\mathrm{loc}}\right)$$
$$\frac{\partial H}{\partial \tau} - \frac{\chi}{L}\frac{dL}{d\tau}\frac{\partial H}{\partial \chi} + \frac{1}{WL}\frac{\partial}{\partial \chi}(UHW) = \dot{B}. \tag{A6}$$

The granular fluidity depends on $\dot{\epsilon}_e$, which is transformed as

$$\dot{\epsilon}_e = \frac{\dot{\epsilon}_\chi}{L}, \tag{A7}$$

where $\dot{\epsilon}_\chi$ is the second invariant of the strain rate in the stretched coordinate system. The transverse velocity equation is
unaffected by the coordinate transformation.

**Appendix B: Nondimensionalization**

We nondimensionalize the model equations to improve model convergence. We start by assuming that we know characteristic
scales for the length $[L]$, velocity $[U]$, and mass balance rate $[\dot{B}]$. We then set scales for the thickness and time:

$$[L] = \frac{[U][H]}{[\dot{B}]}$$
$$[\tau] = \frac{[L]}{[U]} \tag{B1}$$



The model is scaled by defining

$$L = [L]L^*$$

$$H = [H]H^*$$

$$U = [U]U^*$$

$$\dot{B} = [\dot{B}]\dot{B}^*$$

$$W = [L]W^*$$

$$d = [H]d^*, \tag{B2}$$

where $^*$ is used to indicate dimensionless variables. We also note that $g' = g'^*[U]/[L]$ since $g' = \dot{\epsilon}_e/\mu$. Dropping the asterisks and defining $\gamma = [H]^2/[L]^2$, the nondimensional stress balance, granular fluidity, transverse velocity, and mass continuity equations become

$$\frac{1}{L^2}\frac{\partial}{\partial\chi}\left(\frac{H^2}{g^x + (\partial U/\partial x)/L}\frac{\partial U}{\partial\chi}\right) = \frac{H}{L}\frac{\partial H}{\partial\chi} + \frac{H^2}{W}\mu_w\,\mathrm{sgn}(U)$$

$$\frac{\gamma}{L^2}\frac{\partial^2 g^x}{\partial\chi^2} = \frac{1}{\xi^2}\left(g^x - g^x_{\mathrm{loc}}\right)$$

$$\gamma\frac{\partial^2 g^y}{\partial y^2} = \frac{1}{\xi^2}\left(g^y - g^y_{\mathrm{loc}}\right)$$

$$U + (U_t - U_c) = \frac{2}{W}\int_0^{W/2}\int_0^y 2\mu_w\left(1 - \frac{2y}{W}\right)g^y\,dy'\,dy$$

$$\frac{\partial H}{\partial\tau} - \frac{\chi}{L}\frac{dL}{d\tau}\frac{\partial H}{\partial\chi} + \frac{1}{WL}\frac{\partial}{\partial\chi}(UHW) = \dot{B}. \tag{B3}$$

Using dimensionless variables, the cooperativity length and local granular fluidity are calculated as

$$\xi = \frac{Ad}{\sqrt{|\mu - \mu_s|}} \tag{B4}$$

and

$$g^x_{\mathrm{loc}} = g^y_{\mathrm{loc}} = \begin{cases} \dfrac{[L]}{[U]}\sqrt{\dfrac{\tilde{P}}{\rho d^2[H]}}\dfrac{(\mu - \mu_s)}{\mu b} & \text{if } \mu > \mu_s \\ 0 & \text{if } \mu \leq \mu_s. \end{cases} \tag{B5}$$

When calculating $g^x$, the effective coefficient of friction is given by $\mu = (\dot{\epsilon}_\chi/L + \delta\dot{\epsilon})/g^x$, and when calculating $g^y$ it is given by $\mu = \mu_w(1 - 2y/W)$ and $\partial u/\partial y = 2\mu g^y$.

The boundary conditions are unchanged in dimensionless variables.

## Appendix C: Finite difference discretization

We use finite differences with a staggered grid and implicit time step to calculate $U$, $g'_{xx}$, $\mu_w$, $H$, and $L$. Indices $j$ and $n$ refer
to grid points and time steps. We define $j = 0:N$, so that there are $N+1$ grid points each for $U$ and $\mu_w$ and $N$ points each



for $H$ and $g'_{xx}$. Altogether the model solves for $4N+3$ unknowns in the $x$-direction. The discretized stress balance, granular fluidity, $\mu_w$, and mass continuity equations provide $4N+2$ equations. One additional equation comes from defining the end of the ice mélange as being where the thickness equals the grain size:

$$3H_{N-1/2} - H_{N-3/2} = d. \tag{C1}$$

## C1  Stress balance equation

The discretized stress balance equation is

$$\frac{1}{(L\Delta\chi)^2}\left[\nu_{j-1/2}U_{j-1} - \left(\nu_{j+1/2} + \nu_{j-1/2}\right)U_j + \nu_{j+1/2}U_{j+1}\right] =$$

$$\frac{1}{L\Delta\chi}\frac{H_{j+1/2} + H_{j-1/2}}{2}\left(H_{j+1/2} - H_{j-1/2}\right) + \frac{1}{2}\frac{\left(H_{j+1/2} + H_{j-1/2}\right)^2}{W_{j+1/2} + W_{j-1/2}}\mu_{w,j}\operatorname{sgn}(U_j), \tag{C2}$$

which is used for $j = 1 : N - 1$ and where we have defined

$$\nu_{j-1/2} = \frac{H_{j-1/2}^2}{g_{j-1/2}^x + (U_j - U_{j-1})/(L\Delta\chi)}. \tag{C3}$$

The upstream boundary condition is $U_0 = U_c H_t/H_0$ (Equation 20), while the downstream boundary condition is $U_N - U_{N-1} = 0$ (Equation 21).

## C2  Nonlocal granular fluidity equation

The equation for the granular fluidity is discretized using a standard difference formula, such that

$$\gamma\frac{g_{j-3/2}^x - 2g_{j-1/2}^x + g_{j+1/2}^x}{(L\Delta\chi)^2} = \frac{1}{\xi_{j-1/2}^2}\left(g_{j-1/2}^x - g_{\mathrm{loc},j-1/2}^x\right), \tag{C4}$$

with boundary conditions $g_{3/2}^x - g_{1/2}^x = 0$ and $g_{N-1/2}^x - g_{N-3/2}^x = 0$ ($\partial g^x/\partial x = 0$ at $x = 0, L$). The granular fluidity is only calculated on $N$ grid points because it depends on the velocity gradient, which we calculate using a one-sided difference. As result, $g_{j-1/2}^x$ depends on $U_j$ and $U_{j-1}$. Similarly, $g_{\mathrm{loc},j-1/2}^x$ (Equation 15) depends on $U_j$ and $U_{j-1}$ as well as $H_{j-1/2}$.

## C3  Transverse velocity equation

We calculate transverse velocity profiles at each $\chi$ grid point. We use $M + 1$ grid points in the $y$-direction. The discretized granular fluidity equation in the $y$-direction is then

$$\gamma\frac{g_{m-1}^y - 2g_m^y + g_{m+1}^y}{\Delta y^2} = \frac{1}{\xi_m^2}\left(g_m^y - g_{\mathrm{loc},m}^y\right). \tag{C5}$$

At the boundaries we set $dg^y/dy = 0$, and therefore $g_1^y - g_0^y = 0$ and $g_M^y - g_{M-1}^y = 0$. $g_{\mathrm{loc},m}^y$ and $\xi_m$ both depend on $\mu$ (see Equations 15 and 16). For shear-dominated flow, $\mu$ varies linearly across the fjord (Equation 22). Therefore, for a given value of $\mu_w$, $g_{\mathrm{loc},m}^y$ and $\xi_m$ can be directly calculated. Equation (C5) is then solved to determine $g^y(y)$.



Finally, we integrate Equation (23) twice to find the average velocity in the transect, which is required to equal the velocity $U$ in the ice mélange's reference frame plus the glacier terminus velocity:

$$U_j + U_t - U_c = \frac{2}{W_j} \int\limits_0^{W_j/2} \int\limits_0^y \mu_{w,j} \left( 1 - \frac{2y'}{W_j} \right) g^y \, dy' \, dy. \tag{C6}$$

## C4 Mass continuity equation

For the mass continuity equation we use an upwind scheme with a backward Euler step; the advective term is discretized with
centered differences:

$$\frac{H_{j+1/2} - H_{j+1/2}^\star}{\Delta\tau} - \left( \chi_{j+1/2} \frac{dL}{d\tau} \right) \frac{H_{j+3/2} - H_{j-1/2}}{2L\Delta\chi} +$$
$$\frac{1}{W_{j+1/2}} \frac{(U_{j+1} + U_j) H_{j+1/2} W_{j+1/2} - (U_j + U_{j-1}) H_{j-1/2} W_{j-1/2}}{2L\Delta\chi} = \dot{B}_{j+1/2}, \tag{C7}$$

where superscript $^\star$ is now used to refer to values from the previous time step. At both boundaries ($j = 0$ and $j = N - 1$) we use one-sided differences for the advective term, and at the upstream boundary ($j = 0$) we use a forward difference for the diffusive term. Consequently, the discretized mass continuity equations at the upstream and downstream boundaries are

$$\frac{H_{1/2} - H_{1/2}^\star}{\Delta\tau} - \left( \chi_{1/2} \frac{dL}{d\tau} \right) \frac{H_{3/2} - H_{1/2}}{L\Delta\chi} + \frac{(U_2 + U_1) H_{3/2} W_{3/2} - (U_1 + U_0) H_{1/2} W_{1/2}}{2W_{1/2} L\Delta\chi} = \dot{B}_{1/2} \tag{C8}$$

and

$$\frac{H_{N-1/2} - H_{N-1/2}^\star}{\Delta\tau} - \left( \chi_{N-1/2} \frac{dL}{d\tau} \right) \frac{H_{N-1/2} - H_{N-3/2}}{L\Delta\chi} +$$
$$\frac{(U_N + U_{N-1}) H_{N-1/2} W_{N-1/2} - (U_{N-1} + U_{N-2}) H_{N-3/2} W_{N-3/2}}{2W_{N-1/2} L\Delta\chi} = \dot{B}_{N-1/2}. \tag{C9}$$

*Author contributions.* JMA developed the code, conducted the experiments, and wrote the manuscript, with significant help from AAR, JCB, and KN.

*Competing interests.* The authors have no competing interests.

*Acknowledgements.* Funding for this project was provided by US National Science Foundation grants 2025692, 2025764, and 2025795. JMA worked on the project while visiting Aalto University in Espoo, Finland. We thank RH Jackson for discussions that motivated the modeling exercises.



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
