# Peer review of "A quasi-one-dimensional ice mélange flow model based on continuum descriptions of granular materials"

_EGUsphere, 2024_

## Referee Comment (RC1)

**Review**

**Title:** A quasi-one-dimensional ice melange flow model based on continuum descriptions of granular materials

**Reference:** egusphere-2024-297

This work presents a study of the dynamics of an ice melange within the framework of a granular material in the quasi-static regime. The problem is simplified by a double averaging in depth and width. This makes it possible to study the ice melange dynamics in the streamwise direction by leveraging a non local granular rheology. Particularly, the authors explored the sensitivity of the model to physical and geometrical parameters to understand the role of the ice melange on the buttressing force. Here, as I understand it, a significant advantage of implementing the granular fluidity model is that it allows for a more accurate estimation of the ice melange thickness at the calving front. Consequently, this leads to a more precise calculation of the buttressing force. Then, the authors investigated two cases: a steady state where the calving flux equal the flux of icebergs at the outlet of the ice melange and a quasi static state where there is no calving and no flux at the end of the ice melange. In the quasi-static case, the velocity is found to be constant along the fjord and the thickness profile is supposed to be exponential along the fjord. In the steady state, the buttressing force depends on surface and basal melting rate, decreases with fjord width and increases with calving flux. For the latter, it is found that it is because the buttressing force which depends on glacier thickness, increases more rapidly than the force needed to capsize icebergs. It is therefore suggested that glacier with small calving fluxes would be less influenced by buttressing that the glacier with high calving fluxes.

I would like to thank the authors for this interesting paper. Indeed, I truly appreciate the combination of granular physics and glacier modeling, which I believe offers a more nuanced consideration of the mechanisms within the ice melange. This approach could significantly enhance our understanding of the role of ice melange in calving dynamics, as well as the velocity at which icebergs are released from the fjords. The simplified model represents a crucial initial step toward achieving these objectives, and I would encourage the authors to further develop their modeling in the future. Specifically, I suggest exploring the development of a multi-phase flow model to facilitate the incorporation of the effects of seawater and wind while considering the full 3D granular velocity field.

I found the modeling part of the paper to be interesting, well-written, and comprehensive. Therefore, I have only a few minor comments to offer. However, I have more suggestions regarding the sensitivity analysis, which I found to be occasionally briefly discussed or explored. This is understandable given the wide range of scenarios you decided to explore. It seems one must choose between thoroughly exploring all sensitivities of the model or focusing on a specific aspect for a more detailed treatment. Due to the extensive range of parameters you are exploring, I found it sometimes difficult to follow. This is also because figures you are discussing are not mentioned clearly. Therefore my comments are essentially linked to the clarity of the different messages.

**1 Comments on the model**

**1.** Line 60, you define $\sigma_{ij} = R_{ij} - \tilde{p}\delta_{ij}$ with the pressure positive in extension. In Cuffey and Paterson's book, it is rather $R_{ij} = \sigma_{ij} - \tilde{p}\delta_{ij}$ with the same sign convention. Is it a mistake or do you define the overburden pressure differently? Make sure that it is correct and that it doesn't change expressions. In particular, with the formulation in line 60, I cannot recover equation (5).

**2.** How you recover equation (2) from equation (1) is not entirely clear to me. Is it by integrating the equation line 66 with the definition of the gravity of equation (1)? In this case, the statement line 71 should be modified. Then, how do you integrate it? This is related to another comment: I have the feeling that a scheme of the problem considered with characteristic parameters could help the reader to figure out the configuration more directly. For example, line 154 to 158 it could help to visualize the boundary conditions etc...This is just a suggestion.

**3.** Equation (22) is not obvious to me. Could you please cite or explain from where it comes from?

**4.** From what I understand, $\mu_w$ is here an effective friction coefficient and not the usual threshold for movement, which depends on the properties of the material in contact. Is it correct? Then, it is not very clear how you calculate $\mu_w$. From my understanding, you calculate the granular fluidity $g^y$ to solve the averaged transverse velocity field of equation (24) and obtained by averaging equation (23) over the width. But how do you find $\mu_w$ since it is not a constant parameter?

**2 Comments on the analysis**

**5.** Line 231 you stated $dL/dt = 0$ with a full derivative. What is the reason? What are the parameters that affects L? Is it only the ice thickness?

**6.** Line 238 to 246 I really like the discussion but you should may be refer the reader to the different subfigures you are analysing. It would make the analysis straightforward to understand. Same for line 262, you should cite figure 2b.

**7.** Line 244, the "roughly exponential thickness" for the quasi-static limit is not obvious to me. Why is it so different from the steady state?

**8.** In the sensitivity analysis I have the feeling that you clould sometimes discuss more the behavior of the ice melange in light of the physical model. It could allow one to better identify the processes at play. For example, Line 267, you say: " Increasing b makes the ice mélange more stiff and extensive". First I am not sure to understand what you intend by "stiff and extensive". Then, you could may be discuss a bit more why is that so? Ii it because b diminishes the local source term in the diffusion equation of the granular fluidity (equation 14), therefore there is less capacity to diffuse fluidity in the melange?

**9.** Line 277, it is mentioned that ice melange geometries and velocity profiles appear to be roughly consistent with field observations. In figures 2a to 2e, would it be of interest to add a profile from field observations allowing for a rapid comparison? I agree that it is not easy to determine the best calibration as there are different possible combinations of values for the parameters to recover field observations. It would require a deeper analysis that seems to go beyond the purpose of this work.

**10.** I found section **3.1.2 Sensitivity to external forcings and fjord geometry** difficult to follow. In line 284 you say that you will investigate the impact of calving fluxes on ice melange flow and geometry, but then, in line 288 it seems that you do the contrary: you investigate how ice melange plays a role on calving fluxes and start to study the buttressing. Therefore you never discussed clearly your results in light of the different panels of figure 3. Also, I found the discussion from line 285 to 296 sometimes difficult to understand. Here are the main reasons:

- line 286: "The ice mélange becomes more extensive as the fluxes increase (Figure 3), implying that ice mélange produced by highly active glaciers is more likely to exert high resistive stresses against the glacier termini and to persist year round." By extensive you mean longer or thicker? From my understanding of equation (28) only ice melange thickness plays a role. But on the contrary, to me the word "extensive" suggests longer but not thicker.

- line 292, it is not clear how you find these percentages? It increases compared to what?

- The issue linked with icebergs that capsize is not very clear. Is it because once capsized, an iceberg generates thinner ice melange?

I would suggest to rewrite this section more clearly to highlight the messages and be better articulated with the study of the butressing in section 3.2 and 3.3. I have the feeling that it would make the end of the paper more straightforward to follow.

**11.** In the conclusion, line 361, you say that the NSMA produces realistic thickness and velocity profiles. I do not doubt about it but you never showed profiles from the field for a qualitative comparison. This is related to comment 9.

**3   Minor comments**

**1.** I really like that you mention the estimated value for the inertial number. I do not know if this number is largely known in the glaciological community. May be you could define it rapidly and cite work like Gdr Midi 2004?

**2.** Line 253,would it be clearer to say "explore how adjusting these parameters affects the steady-state"?

**3.** Line 262, please cite the figure that allows you to say that "the size of iceberg allows the ice melange to thin and advance". This will be easier to follow the argument.

**4.** You should may be identify the different cases (small glacier, medium glacier and large glacier) in the caption of figure 3. I suppose the first case is for the small glacier, then the medium glacier and lastly the large glacier, but it would make the identification straightforward.

**5** Suggestion: in the appendix, you could may be provide a scheme for the successive steps of your algorithm?

---

## Referee Comment (RC2)

**Review egusphere-2024-297**

**A quasi-one-dimensional ice mélange flow model**
**based on continuum descriptions of granular materials**

**1 General comments**

This paper derives a novel continuum-mechanical model for ice melange dynamics. In recent years, ice melange has been recognized to buttress glacier flow and affect calving fluxes, among other things, which makes it important to include in ice sheet models. However, its properties resemble those of a granular medium, which makes its representation in conventional ice sheet models challenging. This study provides an important step towards including these effects.

My comments below focus mostly on clarifying the derivation of the model, which I think is the most important part of the paper.

**2 Specific comments**

a) Line 55: Before jumping right into the derivation of the model, it would be very helpful for the readers if the authors briefly summarized the assumptions upon which the model is going to be based and where the main differences to the traditional SSA model are going to be. Providing these pointers early will help the reader to understand what is going on later.

b) I would also suggest a sketch of the geometry. Especially later when the integrations are performed, readers are expected to be familiar with the vertical integration of momentum balance equations and which boundary conditions are applied, which is probably not true for everyone.

c) The model is stated incompletely: please write out which boundary conditions are assumed, that is, where is the surface of the melange, where is the base, what is the inflow boundary condition, what is the outflow, what are the stresses/velocities there, this sort of thing.

d) Line 59: "Tectonic stress" – I hadn't come across this term before. In the continuum-mechanical literature usually deviatoric stresses are used which are defined in a similar fashion — also in your equation (5) below. Could you add a short note here explaining how these stresses differ from deviatoric stresses and why you use them?

e) Generally, there is really no need to introduce strain rates and velocities until after equation (7), which is a more natural place to start talking about how stresses are related to strain rates. Consider re-ordering for clarity.

f) Line 64: "inertial number": can you provide the definition here? Most readers will probably not be familiar with this number

g) Line 66: please typeset $\partial \sigma_{ij}/\partial x_j = \rho g_{\text{eff}} \delta_{iz}$ as a separate equation – this makes it easier for people scanning the text without reading every word to follow the maths

h) Line 71: Integrating from where to where (boundary conditions!)?

i) Line 59 & equation (3): $R_{ij}$ seem to refer to both vertically-averaged and non-vertically averaged stresses. Maybe change the latter to $r_{ij}$?

j) $g'$ is a slightly confusing choice for the granular fluidity, as $g$ is also standard gravity and $g'$ is sometimes used for $(1 - \rho_i/\rho_w)g$. Could you use a different letter?

k) equation (17): is it possible for the denominator to go to zero or become negative? Are you applying a regularization here when solving numerically?

l) line 153: this is a good point to mention that you consider your analysis in a moving reference frame that is fixed at the glacier terminus

m) line 164-165: where exactly is the regularization applied?

n) line 169: "The value of $\mu_w$ is related" $\rightarrow$ "The value of $\mu_w$ in (19) is related"

o) from the text it wasn't clear to me whether $\mu_w$ in equation (19) must come from solution of the $y$–component of the fluidity equation which makes solution of (19) more onerous?

p) Line 284-296: I couldn't quite follow this argument, and I don't think the results in questions are shown in the paper. Maybe add a sketch to explain your argument?

q) Line 297-301: Point out somewhere here that melting enters into the model through $\dot{B}$ in equation (25)

r) Line 298-300: "[...] melange extent is sensitive to melt due to its indirect effect on lateral shear stresses": would that be the second term on the right hand side of equation (19) which depends on $H^2$ (please add reference in manuscript)? If not, how does this dependence come about?

---

## Author Comment (AC1)

**Reviewer #2**

*1 General comments*

This paper derives a novel continuum-mechanical model for ice melange dynamics. In recent years, ice melange has been recognized to buttress glacier flow and affect calving fluxes, among other things, which makes it important to include in ice sheet models. However, its properties resemble those of a granular medium, which makes its representation in conventional ice sheet models challenging. This study provides an important step towards including these effects. My comments below focus mostly on clarifying the derivation of the model, which I think is the most important part of the paper.

Thank you for your comments, which have improved the clarity of the paper.

*2 Specific comments*

a) Line 55: Before jumping right into the derivation of the model, it would be very helpful for the readers if the authors briefly summarized the assumptions upon which the model is going to be based and where the main differences to the traditional SSA model are going to be. Providing these pointers early will help the reader to understand what is going on later.

Thank you. We have added a couple of paragraphs to the beginning of Section 2 as suggested.

b) I would also suggest a sketch of the geometry. Especially later when the integrations are performed, readers are expected to be familiar with the vertical integration of momentum balance equations and which boundary conditions are applied, which is probably not true for everyone.

We have added a figure, as also suggested by Reviewer #1.

c) The model is stated incompletely: please write out which boundary conditions are assumed, that is, where is the surface of the melange, where is the base, what is the inflow boundary condition, what is the outflow, what are the stresses/velocities there, this sort of thing.

All of these things are described in Sections 2.1 and 2.2. However, to improve clarity we included a schematic and also added two paragraphs to the beginning of Section 2 in order to provide some model background prior to getting into the details.

d) Line 59: "Tectonic stress" – I hadn't come across this term before. In the continuum-mechanical literature usually deviatoric stresses are used which are defined in a similar fashion — also in your equation (5) below. Could you add a short note here explaining how these stresses differ from deviatoric stresses and why you use them?

Tectonic (i.e., resistive) stresses are commonly used in glaciology to simplify the depth-integration. We now state that and reference van der Veen and Whillans (1989).

e) Generally, there is really no need to introduce strain rates and velocities until after equation (7), which is a more natural place to start talking about how stresses are related to strain rates. Consider re-ordering for clarity.

We prefer to leave this as is in order to reduce the number of places where we have to introduce more variables. This way we are laying out some of the basic variables in the model before getting into the derivation.

f) Line 64: "inertial number": can you provide the definition here? Most readers will probably not be familiar with this number

Done (also in response to Reviewer #1).

g) Line 66: please typeset $\partial \sigma_{ij} / \partial x_j = \rho g_{eff} \delta_{iz}$ as a separate equation – this makes it easier for people scanning the text without reading every word to follow the maths

Done.

h) Line 71: Integrating from where to where (boundary conditions!)?

From $z$ to the surface, now stated in the text.

i) Line 59 & equation (3): $R_{ij}$ seem to refer to both vertically-averaged and non-vertically averaged stresses. Maybe change the latter to $r_{ij}$ ?

Due to our model assumptions, there are no vertical gradients in flow. The only terms that vary with depth are the static pressure, isometric pressure, and full stress tensor (i.e., not tectonic or deviatoric stresses). We have revised the text to make this more clear, and included bars over the stress tensor when specifically talking about depth-averaged values. We have already used different variables (lowercase vs uppercase) when describing the non-vertically averaged and vertically averaged pressures.

j) $g'$ is a slightly confusing choice for the granular fluidity, as g is also standard gravity and $g'$ is sometimes used for $(1 − \rho i/\rho w)g$. Could you use a different letter?

*We understand the possible confusion, but $g'$ is used by the granular mechanics literature to describe the granular fluidity, and we prefer to be consistent with those studies.*

k) equation (17): is it possible for the denominator to go to zero or become negative? Are you applying a regularization here when solving numerically?

*Yes, this is correct. We now state here that the NSMA requires a regularization. We later discuss our regularization scheme, which ensures that the granular fluidity does not go to 0 (see Section 2.2).*

l) line 153: this is a good point to mention that you consider your analysis in a moving reference frame that is fixed at the glacier terminus

*Done. We also now explain this in the new introductory paragraphs at the top of Section 2.*

m) line 164-165: where exactly is the regularization applied?

*It is applied when inserting μ, which depends on $\varepsilon_e$, into the granular fluidity equations. We added a sentence explaining this.*

n) line 169: "The value of $\mu_w$ is related" → "The value of $\mu_w$ in (19) is related"

*Done.*

o) from the text it wasn't clear to me whether μw in equation (19) must come from solution of the y–component of the fluidity equation which makes solution of (19) more onerous?

*It comes from consideration of the shear-dominated regime, so yes, it requires solving the transverse velocity profiles. But really the way to think of this is that we have a system of coupled equations. The longitudinal stress balance equation depends on U and mu_w, and the transverse velocity profiles depend on U and $\mu_w$. Both equations have to be solved at the same time. See beginning of Section 2.3.*

We have also added a couple of introductory paragraphs to Section 2, which we hope will clarify the model set-up and the equations that we solve, including the calculation of $\mu_w$.

p) Line 284-296: I couldn't quite follow this argument, and I don't think the results in questions are shown in the paper. Maybe add a sketch to explain your argument?

This paragraph was revised in response to comments by Reviewer #1. Essentially, increasing the ice thickness (and associated fluxes) from 600 to 800 m produced a 100% increase in the buttressing force. The force required to keep a tall iceberg from capsizing, which can be used to estimate the force required to prevent large-scale calving events, scales with the thickness squared. Increasing the thickness from 600 to 800 m would require a 77% increase in buttressing force to prevent iceberg capsize of full-glacier-thickness icebergs. Thus, the modeled buttressing force increases more rapidly with glacier thickness than the force that would be required to prevent capsize.

q) Line 297-301: Point out somewhere here that melting enters into the model through $\dot{B}$ in equation (25)

Thank you for this suggestion. At the beginning of Section 3.1.2 we now indicate how each of the forcings/parameters enter into the model equations.

r) Line 298-300: "[...] melange extent is sensitive to melt due to its indirect effect on lateral shear stresses": would that be the second term on the right hand side of equation (19) which depends on $H^2$ (please add reference in manuscript)? If not, how does this dependence come about?

Essentially yes, but due to the model's complexity we cannot show this analytically and prefer to leave the text as is. As you point out, that term describes the lateral shear stress and it does depend on $H^2$. However, the shear stress also depends on $\mu_w$, which is not constant (except in the quasi-static limit). For example, for the same calving flux, thinner ice mélange results in faster velocities (e.g., see Equation 22), and faster velocities result in higher values of $\mu_w$.

---

## Author Comment (AC2)

**Reviewer #1**

This work presents a study of the dynamics of an ice melange within the framework of a granular material in the quasi-static regime. The problem is simplified by a double averaging in depth and width. This makes it possible to study the ice melange dynamics in the streamwise direction by leveraging a non local granular rheology. Particularly, the authors explored the sensitivity of the model to physical and geometrical parameters to understand the role of the ice melange on the buttressing force. Here, as I understand it, a significant advantage of implementing the granular fluidity model is that it allows for a more accurate estimation of the ice melange thickness at the calving front. Consequently, this leads to a more precise calculation of the buttressing force. Then, the authors investigated two cases: a steady state where the calving flux equals the flux of icebergs at the outlet of the ice melange and a quasi static state where there is no calving and no flux at the end of the ice melange. In the quasi-static case, the velocity is found to be constant along the fjord and the thickness profile is supposed to be exponential along the fjord. In the steady state, the buttressing force depends on surface and basal melting rate, decreases with fjord width and increases with calving flux. For the latter, it is found that it is because the buttressing force, which depends on glacier thickness, increases more rapidly than the force needed to capsize icebergs. It is therefore suggested that glaciers with small calving fluxes would be less influenced by buttressing than the glacier with high calving fluxes.

I would like to thank the authors for this interesting paper. Indeed, I truly appreciate the combination of granular physics and glacier modeling, which I believe offers a more nuanced consideration of the mechanisms within the ice melange. This approach could significantly enhance our understanding of the role of ice melange in calving dynamics, as well as the velocity at which icebergs are released from the fjords. The simplified model represents a crucial initial step toward achieving these objectives, and I would encourage the authors to further develop their modeling in the future. Specifically, I suggest exploring the development of a multi-phase flow model to facilitate the incorporation of the effects of seawater and wind while considering the full 3D granular velocity field.

I found the modeling part of the paper to be interesting, well-written, and comprehensive. Therefore, I have only a few minor comments to offer. However, I have more suggestions regarding the sensitivity analysis, which I found to be occasionally briefly discussed or explored. This is understandable given the wide range of scenarios you decided to explore. It seems one must choose between thoroughly exploring all sensitivities of the model or focusing on a specific aspect for a more detailed treatment. Due to the extensive range of parameters you are exploring, I found it sometimes difficult to follow. This is also because figures you are discussing are not mentioned clearly. Therefore my comments are essentially linked to the clarity of the different messages.

We would like to thank the reviewer for their encouraging comments and thorough and thoughtful review. We also agree with the reviewer regarding future model developments.

*1 Comments on the model*

1. Line 60, you define $\sigma_{ij} = R_{ij} - \tilde{p}\delta_{ij}$ with the pressure positive in extension. In Cuffey and Paterson's book, it is rather $R_{ij} = \sigma_{ij} - \tilde{p}\delta_{ij}$ with the same sign convention. Is it a mistake or do you define the overburden pressure differently? Make sure that it is correct and that it doesn't change expressions. In particular, with the formulation in line 60, I cannot recover equation (5).

Thank you for this comment. I think the confusion here is related to whether we are discussing resistive (tectonic) stresses or deviatoric stresses. With regards to resistive stresses, our definition is consistent with both Cuffey and Paterson (equations 8.48) and van der Veen (equations 3.14 and 3.15). If this was not the case then we would not arrive at the same vertically integrated stress balance equations. However, you are correct that there was a negative sign error in equation 5. In troubleshooting, we discovered another negative sign error that essentially canceled out the error in equation 5. Following the common convention in glaciology, we had defined the isometric pressure as P = tr(sigma_ij)/3. However, the granular mechanics literature, from which our rheology comes, defines P = -tr(sigma_ij)/3. This means that our definition of pressure was inconsistent with the granular rheology. Instead of modifying the rheology, we now adopt the latter definition of pressure and corrected the negative sign error in equation 5. The upshot is that the model equations remain unchanged; there were just some minor changes to the text to correct the derivation.

In addition, we added a couple of equations (see equations 9-11) to better describe how we arrived at the final rheology.

2. How you recover equation (2) from equation (1) is not entirely clear to me. Is it by integrating the equation line 66 with the definition of the gravity of equation (1)? In this case, the statement line 71 should be modified. Then, how do you integrate it? This is related to another comment: I have the feeling that a scheme of the problem considered with characteristic parameters could help the reader to figure out the configuration more directly. For example, line 154 to 158 it could help to visualize the boundary conditions etc...This is just a suggestion.

First, there was a typo in equation 1 that has now been fixed. The integration is then pretty straightforward, since dp/dz = -rho*g_eff and the granular pressure is 0 at the top and bottom of the ice melange. We have added a sentence to clarify this.

We also added a schematic and couple of paragraphs to the beginning of Section 2 to help orient the reader, as also suggested by the other reviewer.

3. Equation (22) is not obvious to me. Could you please cite or explain from where it comes from?

Equation (22) (now equation 23) arises because the stress has to vary linearly across the fjord if there are no longitudinal deviatoric stresses (which is the case for shear-dominated flow). We added a sentence that described how to arrive at this equation and refer the reader to Amundson and Burton (2018), where we also derived the expression. Note that this is similar to the 1D shallow shelf approximation, in which the shear stress also varies linearly across the glacier.

4. From what I understand, $\mu_w$ is here an effective friction coefficient and not the usual threshold for movement, which depends on the properties of the material in contact. Is it correct? Then, it is not very clear how you calculate $\mu_w$. From my understanding, you calculate the granular fluidity $g_y$ to solve the averaged transverse velocity field of equation (24) and obtained by averaging equation (23) over the width. But how do you find $\mu_w$ since it is not a constant parameter?

That is correct: $\mu_w$ is the effective coefficient of friction along the fjord walls and not a classic coefficient of friction like you might encounter in introductory physics. It is the same coefficient that appears in the granular rheology, just evaluated along the fjord walls. We solve for $\mu_w$ as part of our minimization procedure. At the beginning of Section 2 and also in Section 2.3 we discuss that we simultaneously solve for five variables (velocity, granular fluidity, $\mu_w$, thickness, and length). Most of the equations that we are solving contain two or more of these variables. Using updated equation numbers, Equation (25) contains the velocity and mu_w, equation (20) contains the velocity, granular fluidity, and thickness, … So essentially we have a system of coupled equations that we have to solve simultaneously. Note that this is also discussed at the top of Appendix C.

*2 Comments on the analysis*
5. Line 231 you stated dL/dt = 0 with a full derivative. What is the reason? What are the parameters that affects L? Is it only the ice thickness?

The length only varies with time (as a result of the evolution of the ice mélange thickness profile), so it doesn't make sense to write it as a partial derivative.

6. Line 238 to 246 I really like the discussion but you should may be refer the reader to the different subfigures you are analysing. It would make the analysis straightforward to understand. Same for line 262, you should cite figure 2b.

We considered adding references to each of the panels in Figure 1, but decided that it would be too cumbersome since we would have to refer to a different panel after every half-sentence or so. However, we did add two parenthetical remarks to indicate that the solid lines refer to the steady-state limit and dashed lines to the quasi-static limit.

Regarding line 262: We have added references to the various figure panels of Figure 2 throughout this section.

7. Line 244, the "roughly exponential thickness" for the quasi-static limit is not obvious to me. Why is it so different from the steady state?

We have tried to clarify this by pointing out the profile is different in the quasi-static limit because shear stresses are lower (which leads to thinning) and there is no melting (which allows the ice mélange to lengthen).

8. In the sensitivity analysis I have the feeling that you could sometimes discuss more the behavior of the ice melange in light of the physical model. It could allow one to better identify the processes at play. For example, Line 267, you say: " Increasing b makes the ice m´elange more stiff and extensive". First I am not sure to understand what you intend by "stiff and extensive". Then, you could may be discuss a bit more why is that so? Ii it because b diminishes the local source term in the diffusion equation of the granular fluidity (equation 14), therefore there is less capacity to diffuse fluidity in the melange?

Thank you for this suggestion. We have now done this for each of the four parameters. Essentially, the parameters affect the local granular fluidity, which is the fluidity in the absence of flow gradients ($b$, $\mu_s$), the cooperativity length, which essentially describes a stress coupling length ($A$), or both the local granular fluidity and the cooperativity length ($d$).

9. Line 277, it is mentioned that ice melange geometries and velocity profiles appear to be roughly consistent with field observations. In figures 2a to 2e, would it be of interest to add a profile from field observations allowing for a rapid comparison? I agree that it is not easy to determine the best calibration as there are different possible combinations of values for the parameters to recover field observations. It would require a deeper analysis that seems to go beyond the purpose of this work.

We agree that this is a good point, but instead of trying to plot field observations on top of our model results we instead refer the reader to data that is presented in Amundson and Burton (2018), Bevan et al. (2019), and Xie et al. (2019). Simply plotting existing data on top of our model results is not straightforward because we would have to adjust the model geometry and forcings to produce profiles that have correct length and thickness. The paper is already very involved and we are hesitant to make it even more complex; we simply want to indicate that the modeled profiles are reasonable.

10. I found section 3.1.2 Sensitivity to external forcings and fjord geometry difficult to follow. In

line 284 you say that you will investigate the impact of calving fluxes on ice melange flow and geometry, but then, in line 288 it seems that you do the contrary: you investigate how ice melange plays a role on calving fluxes and start to study the buttressing. Therefore you never discussed clearly your results in light of the different panels of figure 3. Also, I found the discussion from line 285 to 296 sometimes difficult to understand. Here are the main reasons:

This section explores both the impacts of external forcings on ice mélange flow and geometry as well as the feedbacks on calving. We see now that this wasn't clearly articulated. We have renamed the section "Sensitivity of ice mélange flow, geometry, and buttressing force to external forcings and fjord geometry" and revised the first sentence in the section accordingly. We also added some language throughout this section to improve the transitions between paragraphs.

• line 286: "The ice mélange becomes more extensive as the fluxes increase (Figure 3), implying that ice mélange produced by highly active glaciers is more likely to exert high resistive stresses against the glacier termini and to persist year round." By extensive you mean longer or thicker? From my understanding of equation (28) only ice melange thickness plays a role. But on the contrary, to me the word "extensive" suggests longer but not thicker.

We mean longer and thicker, and have revised the wording to indicate so. They are connected, since a longer ice mélange will have more resistance to flow from side drag, leading to thicker ice mélange.

• line 292, it is not clear how you find these percentages? It increases compared to what?

The percentages refer to a comparison of the small glacier and large glacier scenarios. This paragraph has been revised accordingly.

• The issue linked with icebergs that capsize is not very clear. Is it because once capsized, an iceberg generates thinner ice melange?

This has been revised for clarity. Basically the idea is that one way to estimate the force required to prevent large-scale calving events is by calculating the buttressing force that would be required to prevent a large iceberg from fully separating away from the glacier terminus (not just fracturing, but also capsizing and pushing off of the glacier).

I would suggest to rewrite this section more clearly to highlight the messages and be better articulated with the study of the butressing in section 3.2 and 3.3. I have the feeling that it would make the end of the paper more straightforward to follow.

See comment above about revising the language in this section to better emphasize that we are also considering the impacts on ice mélange buttressing.

11. In the conclusion, line 361, you say that the NSMA produces realistic thickness and velocity profiles. I do not doubt about it but you never showed profiles from the field for a qualitative comparison. This is related to comment 9.

We have changed this to read that the profiles are roughly consistent with observations, and cite Amundson and Burton (2018), Bevan et al. (2019), and Xie et al. (2019).

*3 Minor comments*
1. I really like that you mention the estimated value for the inertial number. I do not know if this numberis largely known in the glaciological community. May be you could define it rapidly and cite work like Gdr Midi 2004?

In the first paragraph of Section 2.1 we now define the inertial number and reference GDR Midi.

2. Line 253,would it be clearer to say "explore how adjusting these parameters affects the steady-state"?

Changed to "... affects the steady-state model behavior".

3. Line 262, please cite the figure that allows you to say that "the size of iceberg allows the ice melange to thin and advance". This will be easier to follow the argument.

Done.

4. You should may be identify the different cases (small glacier, medium glacier and large glacier) in the caption of figure 3. I suppose the first case is for the small glacier, then the medium glacier and lastly the large glacier, but it would make the identification straightforward.

Done.

5 Suggestion: in the appendix, you could may be provide a scheme for the successive steps of your algorithm?

Although we appreciate this comment in general, it doesn't really make sense for our model because it would just be a single loop. As stated in Section 2.3, we simultaneously solve the

model equations using a fully implicit time step. We have also now reiterated this point in the first sentence of Appendix C.